# Monitoring the Cortical Activity of Children and Adults during Cognitive Task Completion

**DOI:** 10.3390/s21186021

**Published:** 2021-09-08

**Authors:** Marina V. Khramova, Alexander K. Kuc, Vladimir A. Maksimenko, Nikita S. Frolov, Vadim V. Grubov, Semen A. Kurkin, Alexander N. Pisarchik, Natalia N. Shusharina, Alexander A. Fedorov, Alexander E. Hramov

**Affiliations:** 1Baltic Center for Artificial Intelligence and Neurotechnology, Immanuel Kant Baltic Federal University, 236041 Kaliningrad, Russia; hramovamv@info.sgu.ru (M.V.K.); plo@sstu.ru (A.K.K.); v.maksimenko@innopolis.ru (V.A.M.); n.frolov@innopolis.ru (N.S.F.); v.grubov@innopolis.ru (V.V.G.); s.kurkin@innopolis.ru (S.A.K.); alexander.pisarchik@ctb.upm.es (A.N.P.); nnshusharina@gmail.com (N.N.S.); 2Faculty of Computer Science and Information Technology, Saratov State University, 410012 Saratov, Russia; 3Neuroscience and Cognitive Technology Laboratory, Innopolis University, 420500 Kazan, Russia; 4Centro de Tecnología Biomédica, Universidad Politécnica de Madrid, 28223 Madrid, Spain; 5Immanuel Kant Baltic Federal University, 236041 Kaliningrad, Russia; alafedorov@kantiana.ru; 6Department of Theoretical Cybernetics, Saint Petersburg State University, 199034 St. Petersburg, Russia

**Keywords:** Schulte table, EEG sensors, age differences, sensor-level analysis, attentional modulation, cortical activity monitoring, brain–computer interface, human cognitive state, cognitive task

## Abstract

In this paper, we used an EEG system to monitor and analyze the cortical activity of children and adults at a sensor level during cognitive tasks in the form of a Schulte table. This complex cognitive task simultaneously involves several cognitive processes and systems: visual search, working memory, and mental arithmetic. We revealed that adults found numbers on average two times faster than children in the beginning. However, this difference diminished at the end of table completion to 1.8 times. In children, the EEG analysis revealed high parietal alpha-band power at the end of the task. This indicates the shift from procedural strategy to less demanding fact-retrieval. In adults, the frontal beta-band power increased at the end of the task. It reflects enhanced reliance on the top–down mechanisms, cognitive control, or attentional modulation rather than a change in arithmetic strategy. Finally, the alpha-band power of adults exceeded one of the children in the left hemisphere, providing potential evidence for the fact-retrieval strategy. Since the completion of the Schulte table involves a whole set of elementary cognitive functions, the obtained results were essential for developing passive brain–computer interfaces for monitoring and adjusting a human state in the process of learning and solving cognitive tasks of various types.

## 1. Introduction

The nervous system develops through several processes, some of which are complete before birth, while others continue throughout childhood and adolescence into adulthood (see Refs. [1,2,3] for the literature review). White and gray matter show complex patterns of change over the human lifespan [4]. The volume and integrity of white matter gradually increases from childhood to adulthood in many cortical regions [5]. In contrast, gray matter volume increases from infancy through childhood, peaking during adolescence in the frontal, parietal, and temporal areas [6]. Changes in cognitive functions accompany the ongoing development in white and gray matter, including improvements in the intelligence quotient [7], memory [8], attention [9], and executive functions [10].

Neuropsychological and behavioral performance in daily activities mostly relies on the interaction between different cognitive functions rather than their particular aspects. Goal-directed behavior requires sensory information processing and decision making to give a reasonable behavioral response [11]. On the one hand, our decisions depend on the quality of sensory input. On the other hand, they are affected by top–down mechanisms, including context-related expectations and prior knowledge. Thus, substantial evidence or high capacity of the top–down component reduces cognitive demands at the decision-making stage and facilitates the decisions. The evidence accumulation process involves different cognitive functions. For instance, selective attention allows focusing on decision-relevant features and tuning out unimportant details [12]. Cognitive control also aims at the prioritization of relevant over irrelevant information. Working memory maintains these priorities, so that processing resources are allocated with higher priority to the relevant information [13]. Thus, the active involvement of the particular cognitive functions at the earlier stages decreases the cognitive load on the other components at the latter stages. Mechanisms of interaction between the cognitive processes during mental tasks change with age. Uncovering the ways in which they change will substantially complement and advance our knowledge about brain development.

To address this issue, we subjected children and adults to the Schulte table (ST) with the simultaneous recording of their brain electrical signals with EEG sensors and response time. Performing this task relies on several cognitive processes, including visual search, working memory, and mental arithmetic, so the ST is appropriate for the stated goals. Monitoring cortical activity during cognitive task completion with an EEG system allows obtaining objective information about the brain’s functioning and the occurring cognitive processes with a reasonable spatial and good time resolution [14,15,16]. At the same time, electroencephalography is an easy-to-use and safe non-invasive technology that is especially important when working with children.

The applied motivation of the present study is the development of fundamental basics of functioning of passive brain–computer interfaces (BCIs) for monitoring and adjusting a human state in the process of learning and solving cognitive tasks [17]. In this context, it is crucial to reveal age-related changes in cognitive processes and their interactions, allowing to calibrate and optimize BCIs for the corresponding age group. Such studies, in particular, are in demand for neuroeducation [18,19].

## 2. Materials and Methods

### 2.1. Participants

Twenty conditionally healthy volunteers (no diagnosed diseases of the nervous system, no prescribed drugs), non-smokers, right-handed, with normal or corrected-to-normal visual acuity, who were also amateur practitioners of physical exercise participated in the experiment. The volunteers never participated in neurophysiological experiments before. There were two groups of participants: 12 children (9 males, 3 females, aged 7–8) and 10 adults (7 males, 3 females, aged 18–20). For 48 h before the experiment, all subjects were asked to maintain a healthy lifestyle with 8 h of sleep, a limited consumption of alcohol and caffeine, and mild physical activity. The data of two children were excluded from further consideration since one of them misunderstood the task, and the EEG records of the second were too noisy.

The age groups were selected based on the following considerations. Since we pay special attention to the development of BCIs for education, it was essential to select the contrast age groups relevant to the performed task (ST). The first group consisted of second-grade schoolchildren; the task was difficult for them because they were just beginning to master arithmetic operations. On the contrary, the volunteers from the second group, consisting of 1st–2nd-year students of IT specialty from Innopolis University, could efficiently complete the ST since their arithmetic skills were predominantly at a high level.

The participants (and their parents—for children) received instructions about the experiment, including a description of the experimental design, its goals, methods, and potential inconveniences caused by participation. They were able to ask any related questions and were given appropriate answers and clarifications. Each participant filled and signed informed voluntary consent before their participation in the experiment—the adults signed informed consent blank by themselves, while for children, informed consent blank was signed by the parents. All experimental procedures were performed according to the requirements of the Declaration of Helsinki. The design and methodology of the experiment were approved by the Ethics Committee of Innopolis University.

### 2.2. Experimental Task and Related Discussions

The cognitive task was in the form of a Schulte table—a simplified version of the Zahlen–Verbindung-test (ZVT). Similar to ZVT, the ST is the matrix of randomly arranged elements—the size of the matrix is 5×5 in this case, and elements are represented with numbers from “1’’ to “25’’ (Figure 1A). The main difference between the ZVT and the ST is that in the latter case, accomplishing implies finding the numbers in descending order, i.e., 25, 24, ⋯, 2, 1 as quickly as possible. We supposed that performing this task relied on several cognitive processes, including visual search, working memory, and mental arithmetic.

Visual search involves an active scan of the visual environment for a particular object, target, and among other objects, distractors. Accomplishing ST consists of a series of visual search tasks where each number has to be found among the other numbers. The memory task involves the presentation of a material that participants must recall after a certain amount of time or use to reply to an exercise presented to them. To find a number in ST, participants scan the table and memorize the locations of numbers. After several rounds of scanning, they collect more numbers in memory. Thus, in the course of the task, the accomplishing rate becomes dependent on the working memory capacity: the higher the capacity, the higher the processing performance. Mental arithmetic comprises arithmetical calculations using only the brain resources, with no help from supplies, such as pencil and paper, or devices. One of the most robust phenomena in mental arithmetic is the problem size effect, which indicates that the response time increases as the magnitude of the operands in an arithmetic problem increases [20]. For single-digit problems, 2+3 or 8+7, the larger the product of the operation results, then the higher RT is necessary to produce a correct answer [21]. We suppose that subjects perform arithmetical calculations before finding numbers in the ST. For instance, to find numbers 24 and 4, they calculate “25−1” and “5−1”, respectively. Thus, at the beginning of the ST accomplishment, the result of this operation is a two-digit number, whereas at the end of ST, it becomes one-digit.

There is plentiful evidence that these cognitive processes change with age. For instance, the ability to organize a visual search rapidly develops until age twelve [22]. Visual search performance relies on top–down and bottom–up components. The former characterizes the effect of prior knowledge about the target on the rate of its detection. A match between the sensory input and a target determines the detection rate. A bottom–up component represents a match between the local features (e.g., color, orientation) of the target item and sensory input [23]. The 6-to-7-year-old children have a smaller capacity for both top–down and bottom–up components. As a result, they are unable to guide the search appropriately. Moreover, they cannot monitor across the multiple features of a salient target and instead focus on the single local feature [23]. It evidences a development from middle-to-late childhood in both the top–down and bottom–up components of attentional systems used in visual search.

The working memory performance also depends on age. First, the working memory capacity increases during childhood development [24]. Second, to retain the information in working memory, distracting stimuli must be ignored. This critical ability to ignore the distractors also improves during childhood [25]. Working memory capacity is low at a young age, and children are more susceptible to interfering stimuli. In contrast, adults are more accurate during the working memory task and less distractible than children.

Regarding mental calculation, children perform longer operations than adults [26]. Moreover, the problem size affects children and adults in different ways. Usually, a response to small problems is faster and more accurate than to larger ones due to the differences in the strategy [27]. Small problems are solved using fact retrieval. Large problems engage a time-consuming quantity-based procedural strategy, such as counting or decomposing into smaller problems. Rivera et al. [28] reported that younger children recruited more working memory and attentional resources during arithmetic to engage in procedural problem–solving strategies, even for the small addition and subtraction problems. Throughout development, children develop an increasing reliance on fact retrieval and decreasing reliance on procedural strategies [29]. Finally, the mental calculation requires a working memory [30], and children’s mental arithmetic may be constrained by working memory resources rather than their arithmetical competence [31].

To dissociate the effects of different cognitive processes, we divided the ST accomplishment time into three conditions—each of which involved searching eight numbers. We suggested that the arithmetical problem size was unchanged between the first and the second conditions, but decreased in the third condition due to the reduced numerical size of operands and answers. Then, we supposed that the working memory load enhanced in the second condition in contrast to the first condition due to the growing number of items in the memory set array. Finally, we assumed that the amount of distracting information remained constant across conditions since every target was among the twenty-four distractors.

### 2.3. Experimental Procedure

The experimental procedures took place during the first half of the day in a well-lit room. The subject was sitting in a comfortable chair while the Schulte tables were presented on the tablet computer (Figure 1B). The ST had the 10×10 Sm dimensions, and the distance between the table and the subject’s eyes varied in the range of 0.3–0.4 m. The subjects used a stylus to point to the number, enabling us to record the time when each number was pointed. The subjects were instructed not to connect the respective cells by drawing a line, but only point out each found number with a stylus. All participants successively completed five different tables (Figure 1C). During the experiment, a professional psychologist monitored the whole process. The following table was presented 10 s after the previous table had been finished. To characterize the behavioral performance, we introduced the response times $RT_{i}^{j}$ taken to search for the *i*-th number in the *j*-th table. Here, j=1⋯5 represented the table i=24⋯1 corresponded to the number in the table. Thus, $RT_{22}^{1}$ reflected the time spent between the pointing of number “23” and number “22” in the first table (Figure 1D). We did not consider the $RT_{25}^{j}$ since it included the subject’s preparation for the task and exhibited large variation between the subjects.

### 2.4. EEG Recording and Processing

The EEG data were obtained using the electroencephalograph “actiCHamp” (Brain Products, Germany) with the“ActiCap” active Ag/AgCl electrode sensors. Thirty-one EEG channels were arranged on the scalp according to a “10–10” scheme. A ground electrode was put at the forehead in the position of the “Fpz” EEG electrode—the reference electrode situated on the right mastoid. Before electrode mounting, we applied abrasive gel to clean the scalp skin and increase its conductivity. Then, we placed EEG electrodes with the help of conductive gel. The impedances were monitored during the experiment, and their values were <25 kΩ. EEG data were recorded with a sampling of 1000 Hz and filtered. We used a band-pass filter with 0.016 Hz and 100 Hz cut-offs and a notch filter at 50 Hz. EEG contamination caused by electrocardiogram (ECG) and electrooculogram (EOG) was removed via the independent component analysis (ICA). We used the EEGLAB toolbox for MATLAB to apply ICA for the recorded EEG data. Each EEG dataset of 31 channels was decomposed into 31 independent components using the “runica” function. Then, we found the component with the artifact by comparing initial EEG signals (segments with artifacts) with each of the independent components. We removed the component with artifacts by using the “Remove component” tool.

The EEG signals can also be affected by muscle artifacts. It is known that muscle artifact contamination can obscure some important rhythms on EEG such as alpha or beta, so advanced methods for recognizing and eliminating such artifacts were developed [32]. However, these methods commonly require additional biological signals to be recorded along with EEG, such as electromyogram (EMG) or ECG. In the present study, we aimed to obtain results that would help develop passive brain–computer interfaces, so we tried to keep the experimental recording setup appropriate for such interfaces and as minimalistic as possible. Additionally, muscle activity contamination is notoriously high during intense physical activity such as sports exercises. In our experiment, the subject was sitting calmly at the table in a primarily static pose, so we believe muscle artifacts should not be very pronounced. However, any trials with substantial muscle artifact contamination were rejected and completely removed from the dataset. Moreover, the conducted statistical analysis of the obtained data significantly reduces the effect of artifacts on the revealed effects.

For the spectral analysis, we used the wavelet transform, which has proven itself in the analysis of bioelectric signals and has an optimal ratio of temporal and frequency resolution [33]. We calculated the wavelet power (WP) in the frequency band of 1–70 Hz using the Morlet wavelet [33]. For wavelet analysis, we used the Fieldtrip toolbox [34]. To compare the wavelet power between children and adults, we normalized the spectra on the mean power in the 1–70 Hz frequency band.

### 2.5. Statistical Analysis

The $RT_{i}^{j}$ were non-normally distributed in both age groups (children and adults) according to the Shapiro–Wilk test. To analyze the $RT_{i}^{j}$ between the different tables and within the table we used the repeated measures ANOVA with the Greenhouse–Geisser correction. The post hoc analysis was performed via the nonparametric Wilcoxon signed-rank test.

To compare the $RT_{i}^{j}$ between the age groups and between (or within) the tables, we used the mixed-design ANOVA. The post hoc analysis was performed using the nonparametric Mann–Whitney U test. To address the multiple comparison problem when comparing $RT_{i}^{j}$ across numbers i=24⋯1, tables j=1⋯5, and three conditions, we used the cluster-based correction with the randomization technique [35].

The values of RT averaged within the conditions were normally distributed across the children and adults. We used the mixed-design ANOVA to analyze their change across the conditions and between the age groups. The post hoc analysis was performed via an independent-samples *t*-test. The equality of variances was tested via Levene’s test. The mean values of the wavelet power in the different frequency bands were also normally distributed across the subjects. Therefore, we used ANOVA to evaluate their change across the conditions.

The topograms of WP were considered in the time-frequency-spatial domain and compared for different experimental conditions with the help of a cluster-based permutation test to overcome the multiple comparisons problem [35]. In the pairwise comparison, critical a α-level was set to 0.05, while in the cluster-level statistics, it was set to 0.025, which corresponded to a 0.05 false alarm rate in a two-sided test. Finally, the minimal number of the elements in the cluster was set to 2, and the number of permutations was equal to 2000 [35,36]. This number of permutations was sufficient since it was empirically found that their increase does not change the results obtained for the considered problem. For topogram comparison, we again used the FieldTrip toolbox [34].

The Section 3 provides a description of the tests that we used and their parameters.

## 3. Results

### 3.1. Response Time

First, we performed a within-subject analysis of the response times, $RT_{i}^{j}$, that the subjects took to find the *i*-th number in the *j*-th Schulte table. We used a repeated measures ANOVA with the number i=24⋯1 and the table, j=1⋯5 as two within-subject factors. Children demonstrated a significant effect of the number: F(5.2, 46.8)=4.38, p=0.002 and an insignificant effect of the table: F(2.3, 21.1)=1.5, p=0.245. The interaction effect, number × table, was insignificant: F(4.8, 43.1)=4.38, p=0.301. Adults demonstrated a significant effect of the number: F(5.7, 51.6)=9.07, p<0.001 and the table: F(4, 36)=5.38, p=0.002. Similarly with the children, the interaction effect, number × table, was insignificant: F(6, 54)=1.51, p=0.191. Due to the insignificant interaction effect in both groups, we concluded that each table was performed in a similar way. Therefore, we further considered $RT_{i}$ as the averaged $RT_{i}^{j}$ across all tables for children and adults.

Then, we compared the response times, $RT^{j}$ (Figure 2A), taken to accomplish the *j*-th Schulte table between the age groups, children vs. adults, via the mixed-design ANOVA with the table j=1⋯5 as the within-subject factor. We found a significant effect of age: F(1, 18)=29.03, p<0.001 evidenced that the adults took less time (M =37.7 s, SD =12.8) to accomplish the table than the children (M =72.3 s, SD =20.1). We also observed the insignificant effect of the table: F(2.6, 47)=1.62, p=0.201 and an insignificant interaction effect, as the age group × RT: F(2.61, 47.03)=1.41, p=0.252. The obtained results manifested that the adults performed all tables faster than children.

Finally, we compared the response times, $RT_{i}$ (Figure 2B), taken to find the *i*-th number in the Schulte table between the age groups via the mixed-design ANOVA with the number, i=24⋯1, as the within-subject factor. We observed a significant effect of the age: F(1, 18)=25.83, p<0.001 and the number: F(23, 41)=8.0, p<0.001. We also found a significant interaction effect, age group × RT: F(23, 41)=1.705, p=0.023. Thus, we reported the different ways of accomplishing the table for children and adults.

The mean RT in three conditions were compared between the age groups via a mixed-design ANOVA (Figure 2C). We found a significant effect of the condition: F(2, 36)=39.98, p<0.0001 and a significant interaction effect, condition × age group: F(2, 36)=8.2, p=0.001. The post hoc analysis with a paired-samples *t*-test revealed that children’s RT in *condition1* (M =3.2 s, SD =0.96) and *condition2* (M =3.6 s, SD =1.06) did not differ: t(9)=2.25, p=0.05, but their RT in the *condition3* (M =2.21 s, SD =0.69) was lower than in *condition2*: t(9)=−5.5, p<0.0001. Adults’ RT in the *condition1* (M =1.68 s, SD =0.36) and *condition2* (M =1.77 s, SD =0.45) did not differ: t(9)=1.07, p=0.312; however, similarly to children, their RT in the *condition3* (M =1.26 s, SD =0.27) was lower than in *condition2*: t(9)=−7.2, p<0.0001. Finally, the children’s RT exceeded the adults’ RT under all conditions (p<0.002).

We analyzed how the RT changed within the conditions by contrasting it at the beginning and the end of each condition. In the *condition1* (Figure 3A), the effect of the time-moment (begin vs. end) was insignificant: F(1, 18)=0.041, p=0.843. The interaction effect, time-moment × age group was also insignificant: F(1, 18)=0.015, p=0.904. For the *condition2* (Figure 3B), the effect of the time moment was insignificant: F(1, 18)=1.280, p=0.273, as well as the interaction effect: F(1, 18)=0.283, p=0.601. For *condition3* (Figure 3C), we found a significant effect of the time moment: F(1, 18)=18.0, p<0.001, and a significant interaction effect: F(1, 18)=18.0, p=0.008. The post hoc analysis with the paired samples *t*-test revealed that children decreased their RT from 3.89 s (SD =2.19) to 1.02 s (SD =0.33): t(9)=4.01, p=0.003. The adults’ RT decreased from 1.47 s (SD =0.29) to 0.76 s (SD =0.16): t(9)=7.49, p<0.001. The analysis of the pairwise differences showed that all subjects followed the group tendency in both age groups. Finally, the Mann–Whitney U-test revealed that children demonstrated greater RT change (M =2.86 s, SD =2.25) than the adults (M =0.7 s, SD =0.29): Z=2.79, p=0.004 (Figure 3C).

### 3.2. Brain Activity

First, we compared the wavelet power (WP) of children and adults between the three conditions. For both the children and adults, no differences were found between *condition1* and *condition2*. On the contrary, when contrasting children’s WP in *condition3* and *condition2*, we found a significant positive cluster with p=0.037 in the frequency band of 11.25–13.5 Hz. For this frequency band, we averaged the WP and contrasted it between the conditions. As a result, we observed a significant cluster with p=0.006, including EEG sensors in the parietal (P3, P4, Pz) and parieto-central (CP5, CP1, CP2, CP6) regions. The analysis of the pairwise differences revealed that the WP in this cluster was higher under *condition 3* in all children (Figure 4A). Regarding the adults, a significant positive cluster with p=0.037 was observed in the frequency band of 31.75–32.75 Hz. Testing the averaged WP in this band, we observed a significant cluster (p=0.003), including EEG sensors in the frontal (F3, F4, Fz), central (C3, C4, Cz) and fronto-central (FC5, FC1, FC2, FC6) areas. An analysis of the pairwise differences revealed that the WP in this cluster was higher under *condition3* in 10/12 adults (Figure 4B). These results evidenced that neither children nor adults exhibited changes in the WP between *condition1* and *condition2*. Thus, for the further analysis, we averaged WP across these conditions.

The WP of adults was contrasted to that of children under *conditions1,2* and *condition3*. The children demonstrated a higher WP across all frequencies and conditions due to the higher amplitude of EEG signals. Thus, we compared the normalized wavelet power (NWP) in children and adults (see methods). Under *conditions1,2*, we found two positive clusters with p=0.007 and p=0.0182 in the frequency ranges of 8.5–20.75 Hz and 56.25–70 Hz. Based on the obtained results, we defined the frequency bands of interest as α: 8.25–12 Hz, $\beta_{1}$: 12–21.5 Hz, and γ: 56.25–70 Hz. The mean NWP in these bands was compared between the adults and children. In the α-band, a significant cluster (p=0.0002) included occipital (O1,O2, Oz), left-lateralized parietal (P7, P3, P4, Pz), sensorimotor (CP5, CP1, CP2, C3, Cz, FC5, FC1, FC2), as well as left temporal (TP9, T7, FT9), and frontal midline (Fz) EEG sensors. The NWP of this α-cluster in adults (M =1.48, SD =0.2) was higher than in children (M =0.96, SD =0.25) (Figure 5A). In the β-band, a significant cluster (p=0.0019) included the left-lateralized parietal (P7, P3, Pz) and parieto-central (CP5, CP1, Cp2) sensors, as well as the bilateral central (C3, C4, Cz) and fronto-central (FC1, FC2) sensors, left temporal (TP9) and right-lateralized frontal (F4, Fz) sensors. The NWP of this β-cluster in adults (M =0.97, SD =0.18) was higher than in children (M =0.64, SD =0.23) (Figure 5B). In the γ-band, a significant cluster (p=0.006) included EEG sensors in the right-lateralized sensorimotor (P4, CP6, C4, FC6) and frontal (F4) areas. The NWP of this γ-cluster in adults (M =0.49, SD =0.14) was higher than in children (M =0.27, SD =0.14) (Figure 5C). Under *condition3*, we only found one positive cluster with p=0.0302 the frequency range of 9.25–12.25 Hz. Testing the averaged NWP in this band, we observed a significant cluster (p=0.0002) including the left-lateralized parietal (P7, P3, Pz), sensorimotor (CP5, CP1, CP2, C3, Cz, FC5, FC1), the frontal (F7, Fz), and the left temporal (TP9, T7, FT9) EEG sensors. The NWP of this α-cluster in adults (M =1.32, SD =0.18) was higher than in children (M =0.92, SD =0.26) (Figure 5D).

## 4. Discussion

### 4.1. Behavioral Results

We demonstrated that adults accomplished ST faster than children. Children and adults took, respectively, 3.5 s and 1.75 s to find numbers 24–17 as well as numbers 16–9. In contrast, finding the remaining numbers, 8–1, took less time for both children (2.2 s) and adults (1.2 s). Finally, we reported that children rapidly enhanced their search speed at the end of the task. For the last eight numbers, the search time decreased by 70% for children and 40% for adults. As we hypothesized in the Introduction, ST completion relies on memory, arithmetic, and visual search abilities. The reaction time analysis reveals that the memory component may not change through the task. If the subjects gradually memorized the locations of numbers when scanning the table, they would increase their search speed for numbers 16–9 compared to the numbers 24–17. In contrast, we reported a similar search speed for 16–9 and 24–17. Additionally, a famous work by Horowitz and Wolfe [37] reported that search efficiency is not impaired even if the scene is continually shuffled while the observer is trying to search through it. Switching from two-digit to one-digit numbers might also affect visual search performance. While the number of distractors was constant throughout the task, the low-level features of the target changed. The single-digit numbers differed from the two-digit ones in terms of physical size and structure. Thus, they better captured the bottom–up attention, decreasing the search time for children and adults. When the numbers became one-digit, the children reduced the search time by 70%, which was significantly higher than in adults. We hypothesized that the arithmetical problem size was the main factor limiting the completion speed of children.

### 4.2. Results of EEG Analysis and Brain Activity

In both children and adults, EEG power remained similar for numbers 24–17 and 16–9. For numbers 8–1, the adults demonstrated higher frontal β-band power, and the children showed higher parietal α-band power. Adults exhibited higher α-band power than children in the left hemisphere, with the maximal difference achieved at the left temporal electrodes through the task. We supposed that the observed behavioral and electrophysiological signs together reflected the arithmetical strategy changes under the decreasing problem size. At the beginning of the task, the subjects performed a subtraction from a two-digit number resulting in a two-digit answer. In contrast, at the end of the task, the operands and the results became single-digit. Using various strategies in problems of various sizes can be seen as a major contributing factor to the problem size effect [38,39]. For small problem solutions, fact retrieval is usually enough. However, for large problems, procedural strategies are more appropriate. According to the review [40], different arithmetic strategies are associated with different electrophysiological signatures. The works [41,42] showed that retrieval strategies are tied to the more extensive synchronization of the θ-rhythm (4–8 Hz) in the left hemisphere. Moreover, the procedural strategies were associated with desynchronizing the lower α-rhythm (8–10 Hz) in the parieto-occipital area.

The recent work [43] suggests that the cortical network responsible for arithmetic processing involves the frontal and parietal regions. Frontal gyri are tied to cognitive activity, used in mental calculation, such as working memory and planning. There are three significant parietal regions: the intraparietal sulcus (IPS), which is associated with magnitude processing of numerals; the superior parietal lobule (SPL), which supports attention processes in the general domain; and the left angular gyrus (AG), which is tied to verbal processing in the general domain and long-term memory information retrieval.

Thus, high parietal α-band power at the end of the task may reflect the increased reliance of children on the fact-retrieval strategy when the problem size is small. Event-related desynchronization (ERD) in the α-band tends to generally increase with task difficulty. Given the strong correlation between the α-band ERD and task difficulty, we assume the α-band ERD would be more pronounced for procedural problems than retrieval problems. In their work [44], De Shedt et al. reported that procedural problems were tied to more pronounced alpha ERD across all cortex, with the most noticeable difference observed bilaterally in parietooccipital regions. The authors concluded that the present α-band ERD result suggests that procedural problems are more demanding than simple retrieval problems and thus require higher cortical activation. They also mentioned that earlier fMRI studies showed more pronounced bilateral parietal activation in complex (vs. simple) arithmetic problems as well as in procedural (vs. retrieval) problems (see Ref. [44]). In contrast, high frontal β-band power in adults might reflect enhanced reliance on the top–down mechanisms, e.g., cognitive control [45] or attentional modulation [46] rather than a change of arithmetical strategy.

Finally, the adults demonstrated high EEG power over the left hemisphere through the task. According to Ref. [41], it is an EEG-biomarker of the retrieval strategy. The maximal difference between the adults and children was observed at the left parieto-temporal electrodes, possibly reflecting the left AG, a core area of the fact-retrieval strategy.

### 4.3. Educational Aspects

Thus, we suggested that the arithmetical problem size affected children more than adults. On the one hand, this effect may be a result of internal mechanisms of brain development. On the other hand, educational aspects may influence the arithmetic skills of children. The child’s thinking undergoes vital changes during the primary school age: from the visual-effective to the visual-figurative and further to the verbal-logical form. Studying mathematics contributes to the development of thinking by forming abilities of abstract representation of the surrounding world. When studying mathematics, children meet the concept of numbers and learn to solve arithmetic problems [47]. This part, traditionally called arithmetic, forms the basis for studying mathematics in primary school.

Halberda et al. [48] emphasized the importance of the sense of the number for the formation and development of mathematical thinking that may become a critical background to grow formal mathematical abilities. Moreover, the association between success in processing numbers and broader mathematical competence may also be moderated by participant age [49]. The main goal of studying arithmetic in primary school is developing computational skills and abilities in children usually brought to automatism. These skills and abilities develop when mastering specific computing techniques that allow a child to consistently perform a system of operations leading to a calculated result. Having computational skills means knowing operations and their order that guaranty the correct and quick solution of arithmetical problem [50].

When accomplishing the Schulte table task, adults operate with two-digit numbers using fact-retrieval. This is an appropriate brought to automatism technique to effectively solve this problem. We hypothesized that the children participating in the experiment do not fully master the appropriate skill to operate with numbers greater than 10. In other words, they used a demanding procedural strategy instead of fact-retrieval. We assume two reasons for this effect. First, children of primary school age barely have enough experience with two-digit numbers, and the corresponding skills have not yet been brought to automatism. Moreover, the Schulte table task is unusual for them. As a result, children utilize high mental efforts for its successful solution. The second possible explanation lies in the differences between teaching two-digit numbers to modern schoolchildren and representatives of the older generation. The adult participants learned arithmetic using the abstract concept of numbers and the corresponding arithmetic operations with them. In contrast, modern educational techniques in primary school used the visual perception of information. Facing two-digit numbers, pupils in Russia are usually offered pictures with objects located ten in each row. Moving to the perception of a two-digit number as a number with digits, pupils begin to operate with ten as a counting unit using the specially designed illustrations. It is believed that this model is visual and easy to understand. However, such clarity can likely slow down the perception of two-digit numbers at the level of abstraction. Consideration of these issues requires further research and analysis with younger schoolchildren.

The main limitation of this study is that we are considering two age groups (albeit representative ones). This limits our findings on age-related changes in cognitive processes to the age range from 7 to 20 years old. Nevertheless, we believe that this period of human life is the most important for improving many cognitive skills. Considering a more diverse set of age groups is significant research that we plan to conduct in the future.

## 5. Conclusions

On the behavioral level, we observed that adults performed the Schulte table faster than children. For both children and adults, the mean RT in the second condition was equal to one in the first condition. Finally, both children and adults reduced their RT within the third condition, but this effect was more significant in children.

On the neural level, we revealed in children the shift from a procedural strategy to a less demanding fact-retrieval in the third condition when operating with one-digit numbers. This conclusion is supported by higher parietal α-band power in the third condition discovered in a within-subject analysis of the EEG spectral power of children. Adults predominantly relied on the top–down mechanisms, cognitive control, or attentional modulation rather than a change in arithmetic strategy in Schulte table performance. This is supported by higher frontal β-band power in the third condition in adults.

Finally, based on the results of a between-subject analysis, we concluded that adults mostly use the fact-retrieval strategy during the whole experiment. The higher left-lateralized α-band power in adults compared to children during all conditions confirms it. We suppose that children experienced difficulties at the beginning when operating with two-digit numbers. For the one-digit numbers, their performance increased and reached adults’ scores.

The obtained fundamental results are essential for advancing our knowledge about brain development, including age-related changes in cognitive processes during mental tasks. Since the completion of the Schulte table involves a whole set of elementary cognitive functions, the revealed effects are essential for developing passive brain–computer interfaces [17,51] for monitoring and adjusting a human state in the process of learning and solving cognitive tasks of various types during educational process. In particular, the identified patterns of neural activity (biomarkers) associated with different strategies for solving tasks are important for developing algorithms that underlie the functioning of BCIs, which adapt to different age groups.

## Figures and Tables

**Figure 1 sensors-21-06021-f001:**
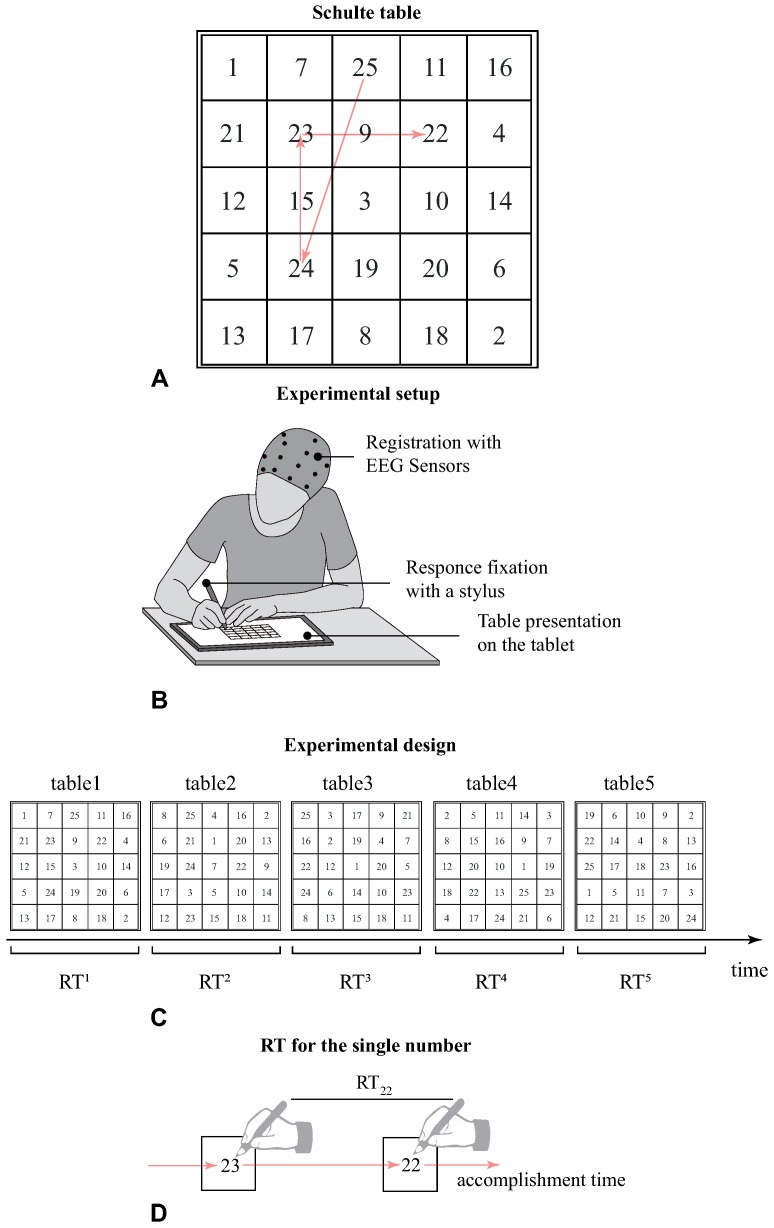
(**A**) An example of the Schulte table. The arrows show the accomplishment process. (**B**) Illustration of the experimental setup. The subject uses a stylus to accomplish the Schulte table on the tablet us screen. (**C**) Experimental design includes accomplishing the five different Schulte tables. (**D**) The schematical illustration of the response time taken to find the number “22” as the time spent between pointing number “22” and pointing the preceding number “23”.

**Figure 2 sensors-21-06021-f002:**
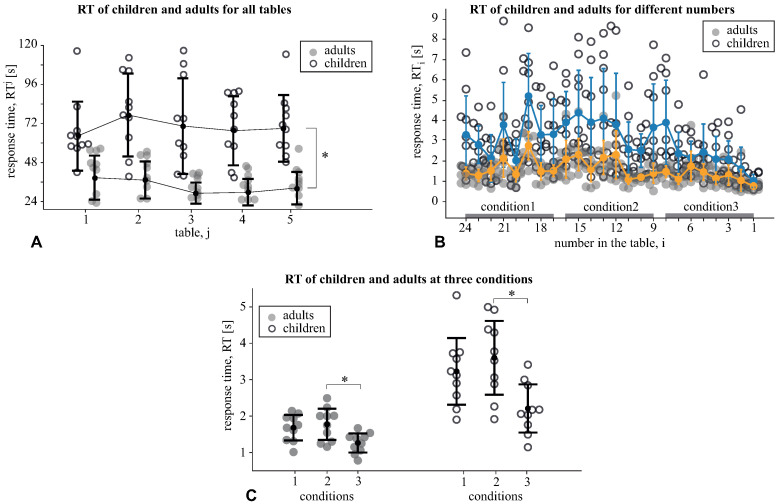
(**A**) The response times $RT^{j}$ of the *j*-th table accomplishment for the children and adults. Data are shown as the mean ± SD and the individual values (${}^{*}p<0.05$ via a mixed-design ANOVA with the GreenHouse–Geisser correction). (**B**) the response time, $RT_{i}$ taken to find the *i*-th number in the Schulte table. Data are shown as mean ± SE for children and adults. Accomplishment time is divided into three intervals marked as *condition1*, *condition2* and *condition3*. Each condition includes searching for eight numbers. (**C**) The mean RT in three conditions for the children and adults (${}^{*}p<0.05$ via a mixed-design ANOVA).

**Figure 3 sensors-21-06021-f003:**
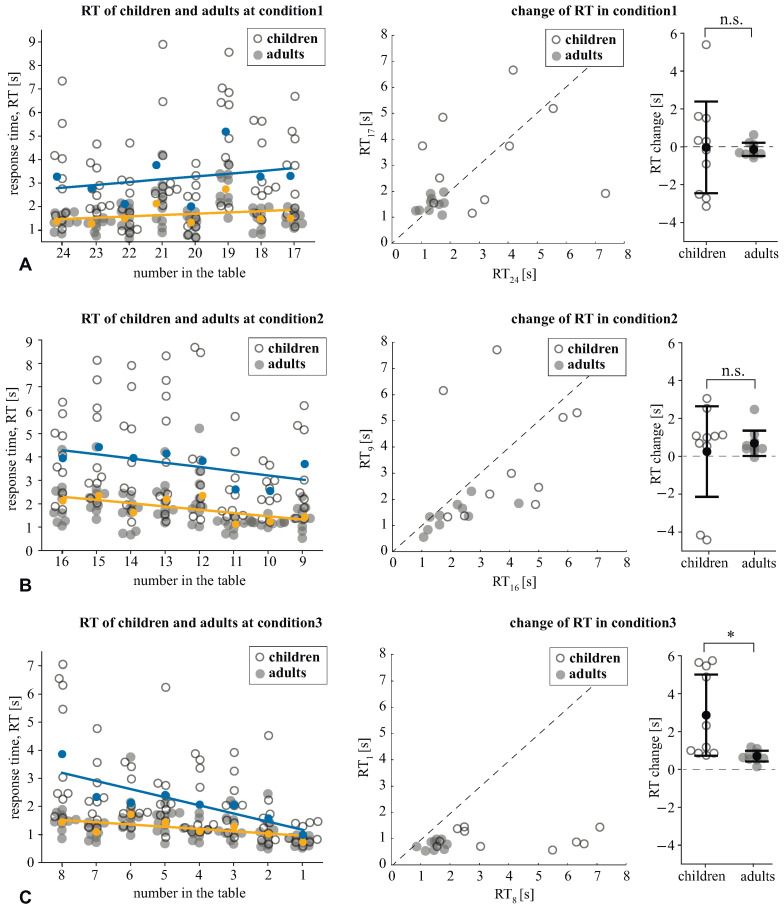
The RT of children and adults under *condition1* (**A**), *condition2* (**B**), and *condition3* (**C**). The left column reflects the RT taken to find each number within the condition. The individual scores of children and adults are shown with their mean values (colored circles). Linear approximation illustrates the way that the mean RT changes within the condition. The middle column shows the individual RTs of children and adults taken to find the first and the last number within the condition. The right column demonstrates the difference between the RT taken to find the last and the first number. Data are shown as the mean ± SD and the individual values for the children and adults (${}^{*}\;p>0.05$ via the Mann–Whitney U-test).

**Figure 4 sensors-21-06021-f004:**
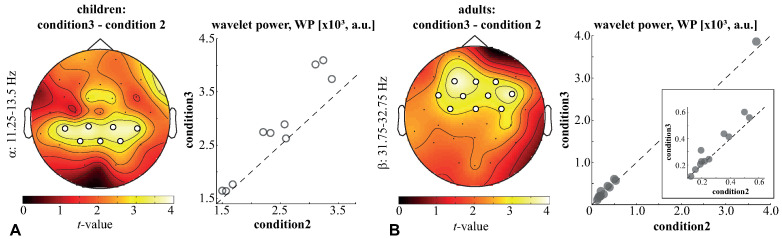
Significant clusters reflect the change of the children’s α-band WP (**A**) and the adults’ β-band WP (**B**) between *condition3* and *condition2*. Topograms illustrate *t*-values, and scatterplots show the pairwise differences of the WP in these clusters between conditions. The frequency ranges determine the positions of the clusters in the frequency domain.

**Figure 5 sensors-21-06021-f005:**
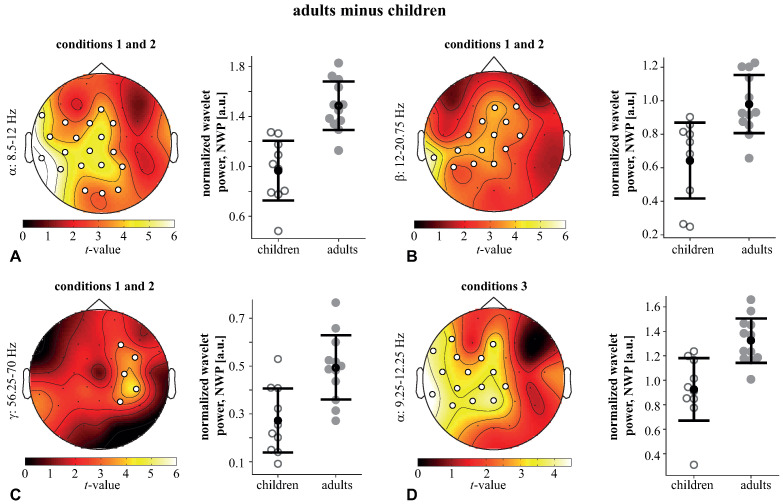
(**A**–**D**) Significant clusters reflect the difference between the adults’ NWP and children’s NWP in the α-, β-, and γ-bands for *conditions1,2*, and in the α-band for *condition3*. Topograms illustrate *t*-values. The NWP in these clusters is also shown as the mean ± SD and individual scores for children and adults.

## Data Availability

The data presented in this study are available on request from the corresponding author. The data are not publicly available due to ethical reasons and the confidentiality of subjects that include children.

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
