# Peer review of "Monitoring the Cortical Activity of Children and Adults during Cognitive Task Completion"

_sensors, 2021, doi:10.3390/s21186021_

Round 1
Reviewer 1 Report
Explanation: The study is not within the scope of Sensors. There is no novel work regarding sensors development/testing done. The clear goal of the study is missing and therefore the transfer to another journal can't be recommended either.
The manuscript presents the analysis of the data collected during completing a cognitive task called Schulte table. Unfortunately, it seems to be focused on the statistical analysis of the collected data without putting the presented research in any scientific or clinical context. There is no research involving sensors development or testing performed which makes the choice of the journal unmotivated. The main part of the introduction is focused on the reported work, while the motivation of the study and background research is missing. Same holds for the discussion part - no comparison of the obtained results with other EEG-focused research of cognitive tasks is provided. There is a mention of potential benefits for the BCI development, but no specific ideas on how exactly those benefits could look like are provided. Why is the age relevant for those potential applications? Further, if we want to perform comparison between children and adults, why would we choose for the age 18-20, when in terms of the brain development subjects are possibly still closer to the group of adolescents than adults?
Minor comments:
The manuscript contains many explanation gaps, which need to be fixed. E.g.:
-explanation of the conditions 1,2,3 appears on fig.1, but not in the main text. The word "condition" also suggests rather externally imposed (experimental) condition (e.g. difficulty level).
-l. 179: "across i=24..1 conditions". Before only 3 conditions are defined, the wording is confusing
-l.187: What are t1,...t8?
-How exactly are WP used in the statistical analysis? How multiple channels are handled? How exactly Morlet wavelet coefficients are related to the frequency bands mentioned in the text?
-Figure 4 with band names appears before any bands are defined. What does it mean e.g. "alpha: 11.25 - 13.5" on the y-axis? Why only this part of the band?
I would suggest very extensive restructuring the paper, focusing on more clear scientific goal, backgrounds, EEG- methods and comparison to the state-of-the art research. The flow of the writing (first define, then use) should be carefully edited.
Author Response
We thank the Reviewer for the critical comments. According to these comments, we have essentially revised the Manuscript, focusing on more clear scientific goals, background, and comparison to state-of-the-art research.
The study is not within the scope of Sensors. There is no novel work regarding sensors development/testing done. The clear goal of the study is missing and therefore the transfer to another journal can’t be recommended either. … There is no research involving sensors development or testing performed which makes the choice of the journal unmotivated.
We do not agree with this statement. The obtained fundamental results and identified patterns of neural activity (biomarkers) are crucial for developing algorithms for handling data from EEG sensors [Hramov, A.E.; Maksimenko, V.A.; Pisarchik, A.N. Physical principles of brain-computer interfaces and their applications for rehabilitation, robotics and control of human brain states. Physics Reports 2021, 918, 1]. Moreover, the revealed effects are essential for developing passive brain-computer interfaces [Jamil, N.; Belkacem, A.N.; Ouhbi, S.; Lakas, A. Noninvasive Electroencephalography Equipment for Assistive, Adaptive, and Rehabilitative Brain–Computer Interfaces: A Systematic Literature Review. SENSORS 2021, 21, 4754; Park, S.; Han, C.H.; Im, C.H. Design of wearable EEG devices specialized for passive brain–computer interface applications. SENSORS 2020, 20, 4572; Zander, T.O.; Kothe, C. Towards passive brain-computer interfaces: applying brain-computer interface technology to human-machine systems in general. Journal of neural engineering 2011, 8, 025005] for monitoring and adjusting a human state in the process of learning and solving cognitive tasks of various types during educational process. In particular, the identified biomarkers associated with different strategies for solving tasks are important for developing algorithms that underlie the functioning of BCIs, which adapt to different age groups.
We have added such clarifications to the Introduction section.
The manuscript presents the analysis of the data collected during completing a cognitive task called Schulte table. Unfortunately, it seems to be focused on the statistical analysis of the collected data without putting the presented research in any scientific or clinical context.
When we deal with EEG and behavioral data in experiment, statistical analysis is critical. It allows one to make significant and mathematically justified conclusions. As a result, we pay special attention to the results of statistical analysis. The presented research corresponds to the following scientific context. The obtained fundamental results are essential for advancing our knowledge about brain development, including age-related changes in cognitive processes during mental tasks. The applied motivation of the present study is the development of fundamental basics of functioning of passive brain-computer interfaces (BCIs) for monitoring and adjusting a human state in the process of learning and solving cognitive tasks [Hramov, A.E.; Maksimenko, V.A.; Pisarchik, A.N. Physical principles of brain-computer interfaces and their applications for rehabilitation, robotics and control of human brain states. Physics Reports 2021, 918, 1]. In this context, it is crucial to reveal age-related changes in cognitive processes and their interaction, allowing calibrating and optimizing BCIs for the corresponding age group. Such studies, in particular, are in demand for neuroeducation [Jamil, N.; Belkacem, A.N.; Ouhbi, S.; Lakas, A. Noninvasive Electroencephalography Equipment for Assistive, Adaptive, and Rehabilitative Brain–Computer Interfaces: A Systematic Literature Review. Sensors 2021, 21, 4754; Park, S.; Han, C.H.; Im, C.H. Design of wearable EEG devices specialized for passive brain–computer interface applications. Sensors 2020, 20, 4572].
The main part of the introduction is focused on the reported work, while the motivation of the study and background research is missing.
We thank the Reviewer for the valuable comment. We have revised the Introduction section.
Same holds for the discussion part - no comparison of the obtained results with other EEG-focused research of cognitive tasks is provided.
We reconsidered the Discussion section. Below are the most important comparisons of the obtained results with other EEG-focused research of cognitive tasks.
As we hypothesized in the introduction, ST completion relies on memory, arithmetics, and visual search abilities. The reaction time analysis reveals that the memory component may not change through the task. If the subjects gradually memorized the locations of numbers when scanning the table, they would increase search speed for the numbers 16-9 compared to the numbers 24-17. In contrast, we reported a similar search speed for 16-9 and 24-17. Additionally, a famous work of Horowitz and Wolfe [Horowitz, T.S.;Wolfe, J.M. Visual search has no memory. Nature 1998, 394, 575–577] reported that search efficiency was not impaired, even if the scene was continually shuffled while the observer was trying to search through it.
We supposed that observed behavioral and electrophysiological signs together reflected arithmetical strategy changes under the decreasing problem size. At the beginning of the task, subjects performed subtraction from a two-digit number resulting in the two-digit answer. In contrast, at the end of the task, the operands and the result became single-digit. The application of different strategies in problems of different sizes is assumed to be a major contributing factor to the problem size effect [Campbell, J.I.; Xue, Q. Cognitive arithmetic across cultures. Journal of Experimental Psychology: General 2001, 130, 299; LeFevre, J.A.; Sadesky, G.S.; Bisanz, J. Selection of procedures in mental addition: Reassessing the problem size effect in adults. Journal of Experimental Psychology: Learning, Memory, and Cognition 1996, 22, 216]. Small problems are usually solved utilizing fact retrieval, whereas large problems are often solved via procedural strategies. According to the review [Hinault, T.; Lemaire, P. What does EEG tell us about arithmetic strategies? A review. International Journal of Psychophysiology 2016, 106, 115], arithmetic strategies have different electrophysiological signatures. Grabner and De Smedt [Grabner, R.H.; De Smedt, B. Neurophysiological evidence for the validity of verbal strategy reports in mental arithmetic. Biological psychology 2011, 87, 128–136; Grabner, R.H.; De Smedt, B. Oscillatory EEG correlates of arithmetic strategies: a training study. Frontiers in psychology 2012, 3, 428] found that retrieval strategies were associated with more extensive synchronization of the theta-band over the left hemisphere of the scalp (4-8 Hz) compared to procedural strategies. Moreover, the procedural strategies were accompanied by parieto-occipital desynchronization of the lower alpha-band (8-10 Hz).
The recent work [Soltanlou, M.; Artemenko, C.; Ehlis, A.C.; Huber, S.; Fallgatter, A.J.; Dresler, T.; Nuerk, H.C. Reduction but no shift in brain activation after arithmetic learning in children: a simultaneous fNIRS-EEG study. Scientific reports 2018, 8, 1] suggests that the cortical network underlying arithmetic processing includes frontal and parietal regions.
Thus, the strong association between alpha-band ERD and task difficulty suggests that procedural problems elicit more alpha-band ERD than retrieval problems. In their work [De Smedt, B.; Grabner, R.H.; Studer, B. Oscillatory EEG correlates of arithmetic strategy use in addition and subtraction. Experimental brain research 2009, 195, 635], De Shedt et al. reported that procedural problems were accompanied by stronger alpha ERD than retrieval problems in all cortical areas, with the largest difference emerging in parietooccipital regions, bilaterally. The authors concluded that the present alpha-band ERD result might reflect higher cortical activation in the more demanding procedural than simple retrieval problems. They also mentioned that previous fMRI studies showed stronger bilateral parietal activation in complex (vs. simple) arithmetic problems as well as in procedural (vs. retrieval) problems. In contrast, high frontal beta-band power in adults might reflect enhanced reliance on the top-down mechanisms, e.g., cognitive control [Stoll, F.M.;Wilson, C.R.; Faraut, M.C.; Vezoli, J.; Knoblauch, K.; Procyk, E. The effects of cognitive control and time on frontal beta oscillations. Cerebral cortex 2016, 26, 1715] or attentional modulation [Lee, J.H.; Whittington, M.A.; Kopell, N.J. Top-down beta rhythms support selective attention via interlaminar interaction: a model. PLoS Comput Biol 2013, 9, e1003164] rather than a change of arithmetical strategy.
Finally, adults demonstrated high EEG power over the left hemisphere through the task. According to Ref. [Grabner, R.H.; De Smedt, B. Neurophysiological evidence for the validity of verbal strategy reports in mental arithmetic. Biological psychology 2011, 87, 128], it is an EEG-biomarker of the retrieval strategy. The maximal difference between the adults and children was observed at the left parieto-temporal electrodes, possibly reflecting left AG, a core area of the fact-retrieval strategy.
There is a mention of potential benefits for the BCI development, but no specific ideas on how exactly those benefits could look like are provided.
We thank the Reviewer. We have added the clarification to the Conclusions section. Since the completion of the Schulte table involves a whole set of elementary cognitive functions, the revealed effects are essential for developing passive brain-computer interfaces for monitoring and adjusting a human state in the process of learning and solving cognitive tasks of various types during educational process. In particular, the identified patterns of neural activity associated with different strategies for solving tasks are important for developing algorithms that underlie the functioning of BCIs, which adapt to different age groups. Thus, the obtained results allow us to propose biomarkers, which will become the basis of a passive BCI for monitoring the subject's state, particularly identifying used task-solving strategies [Hramov, A.E.; Maksimenko, V.A.; Pisarchik, A.N. Physical principles of brain-computer interfaces and their applications for rehabilitation, robotics and control of human brain states. Physics Reports 2021, 918, 1].
Why is the age relevant for those potential applications?
Further, if we want to perform comparison between children and adults, why would we choose for the age 18-20, when in terms of the brain development subjects are possibly still closer to the group of adolescents than adults?
The age groups were selected based on the following considerations. Since we pay special attention to the development of BCIs for education, it was essential to select contrast age groups relevant to the performed task (ST). The first group consists of second-grade schoolchildren; the task is difficult for them because they are just beginning to master arithmetic operations. On the contrary, the volunteers from the second group, consisting of 1st-2nd-year students of IT specialty from Innopolis University, will efficiently complete ST since their arithmetic skills are predominantly formed at a high level.
The child's thinking undergoes vital changes during the primary school age: from the visual-effective to the visual-figurative and further to the verbal-logical form. Studying mathematics contributes to the development of thinking by forming abilities of abstract representation of the surrounding world. When studying mathematics, children meet the concept of numbers and learn to solve arithmetic problems [Matejko, A.A.; Ansari, D. The neural association between arithmetic and basic numerical processing depends on arithmetic problem size and not chronological age. Developmental cognitive neuroscience 2019, 37, 100653].
We have added these clarifications to the Participants and Discussion sections.
Minor comments:
The manuscript contains many explanation gaps, which need to be fixed. E.g.:
- explanation of the conditions 1,2,3 appears on fig.1, but not in the main text. The word "condition" also suggests rather externally imposed (experimental) condition (e.g. difficulty level).
We thank the Reviewer for the comment. We have added necessary information about conditions to the “Experimental procedure” section.
-l. 179: "across i=24..1 conditions". Before only 3 conditions are defined, the wording is confusing
It was the misprint. We corrected it. We thank the Reviewer for the remark.
-l.187: What are t1,...t8?
It was an inappropriate phrase. We meant the following: “The values of RT averaged within the conditions were normally distributed across the children and adults”. Corrected.
-How exactly are WP used in the statistical analysis? How multiple channels are handled? How exactly Morlet wavelet coefficients are related to the frequency bands mentioned in the text?
We handled multichannel EEG data according to work [Maris, E.; Oostenveld, R. Nonparametric statistical testing of EEG-and MEG-data. Journal of neuroscience methods 2007, 164, 177] using cluster-based permutation test. This statistical approach allows one to effectively overcome the multiple comparisons problem: instead of evaluating the difference between the experimental conditions for each of the samples separately, it is estimated using a single test statistic for an entire spatial-spectral grid.
Algorithm for calculating test statistics:
(1) For each sample (points on the spatial-spectral grid), we compare the set of trials in two conditions using a t-test.
(2) Select all samples whose t-value exceeds a certain threshold.
(3) We group the selected samples into related sets (clusters) based on information about the spatial-spectral neighborhood.
(4) Compute statistics at the cluster level by taking the sum of the t-values within the cluster.
(5) Select the largest statistic at the cluster level as the test statistic.
Test statistics were calculated multiple times for different data permutations. The described method allows one to identify clusters in the data.
For wavelet analysis we used complex Morlet wavelet with central frequency Omega_0 = 2Pi. Choice of this value of Omega_0 leads to simple relation between wavelet scale S and frequency F: F = 1/S. Wavelet coefficients were calculated and used to calculate WP in frequency range 1-70 Hz. Then WP was averaged over certain frequency band to obtain WP in this band.
-Figure 4 with band names appears before any bands are defined. What does it mean e.g. "alpha: 11.25 - 13.5" on the y-axis? Why only this part of the band?
The frequency ranges were determined from the statistical cluster-based permutation test results and determine the positions of the clusters in the frequency domain. The frequency bands are indicated in the figures only to clarify where the frequencies of the identified cluster belong (to alpha, beta, or gamma band). We have added the necessary clarification to the figure caption.
Reviewer 2 Report
The paper describes the use of an EEG system to monitor and to analyze the cortical activity of children and adults at a sensor level during cognitive tasks. The results of the experiments are presented. The study is well-prepared and methodologically sound, while the results achieved are interesting. However, the paper needs to be revised to address the comments presented below:
- Extend the abstract to include 1-2 sentences about the naming numerical results of this study.
- What is the technical contribution of this study? Known methodology and workflow were used as available from the EEGLAB tool. The authors should clearly state their novelty (technical and/or methodological) and contribution at the end of the Introduction section.
- The gender distribution of the participants is skewed as much more males than females we included. Why gender diversity was not ensured?
- You removed cardiac signals when preprocessing EEG data. But did you consider EMG (electromyography) signal contamination caused by muscle movements? While EMG contamination is greatest at the periphery of the scalp near the active muscles, even weak contractions can produce EMG that obscures or mimics EEG alpha, mu, or beta rhythms over the entire scalp. Recognition and elimination of this contamination is likely to require elaborate signal filtering, which exceeds simple bandpass filtering used in your study. You should check and discuss “Removal of movement artefact for mobile EEG analysis in sports exercises” on how to remove the movement-based noise from the EEG signal in your experiments.
- The motivation for the used methods must be provided as there are many alternatives for each stage of your workflow available.
- Line 193: present a motivation for using 2000 permutations. An appropriate reference can be used to support.
- Figure 3: the use of floating point numbers to enumerate conditions on the x-axis of the plots is confusing.
- Improve the conclusions, present in-depth insights from your study. Current conclusions are too generic and do not follow from the numerical results of the experiments themselves. Use the main findings of your study to support your claim.
- Check the language. There are some typos and misspellings.
- Explain all abbreviations of their first use.
Author Response
The paper describes the use of an EEG system to monitor and to analyze the cortical activity of children and adults at a sensor level during cognitive tasks. The results of the experiments are presented. The study is well-prepared and methodologically sound, while the results achieved are interesting. However, the paper needs to be revised to address the comments presented below:
We are pleased that the Reviewer found our work “well-prepared and methodologically sound” and “interesting”. We thank the Reviewer for the valuable comments.
Extend the abstract to include 1-2 sentences about the naming numerical results of this study.
We considered this comment and corrected the abstract.
What is the technical contribution of this study? Known methodology and workflow were used as available from the EEGLAB tool. The authors should clearly state their novelty (technical and/or methodological) and contribution at the end of the Introduction section.
We thank the Reviewer for the valuable comment.
The main methodological/technical contribution of our study is the development of fundamental basics of functioning of passive brain-computer interfaces (BCIs) based on the system with multiple EEG sensors for monitoring and adjusting a human state in the process of learning and solving cognitive tasks [Hramov, A.E.; Maksimenko, V.A.; Pisarchik, A.N. Physical principles of brain-computer interfaces and their applications for rehabilitation, robotics and control of human brain states. Physics Reports 2021, 918, 1]. In this context, it is crucial to reveal age-related changes in cognitive processes and their interaction, allowing calibrating and optimizing BCIs for the corresponding age group. In particular, the identified patterns of neural activity associated with different strategies for solving tasks are important for developing algorithms that underlie the functioning of BCIs, which adapt to different age groups. Such studies, in particular, are in demand for neuroeducation [Jamil, N.; Belkacem, A.N.; Ouhbi, S.; Lakas, A. Noninvasive Electroencephalography Equipment for Assistive, Adaptive, and Rehabilitative Brain–Computer Interfaces: A Systematic Literature Review. Sensors 2021, 21, 4754; Park, S.; Han, C.H.; Im, C.H. Design of wearable EEG devices specialized for passive brain–computer interface applications. Sensors 2020, 20, 4572].
We have added these clarifications to the Introduction and Conclusions sections.
The gender distribution of the participants is skewed as much more males than females we included. Why gender diversity was not ensured?
Such skewed gender distribution results from the skewness of the gender distributions at the school class and student group from which volunteers were recruited.
You removed cardiac signals when preprocessing EEG data. But did you consider EMG (electromyography) signal contamination caused by muscle movements? While EMG contamination is greatest at the periphery of the scalp near the active muscles, even weak contractions can produce EMG that obscures or mimics EEG alpha, mu, or beta rhythms over the entire scalp. Recognition and elimination of this contamination is likely to require elaborate signal filtering, which exceeds simple bandpass filtering used in your study. You should check and discuss “Removal of movement artefact for mobile EEG analysis in sports exercises” on how to remove the movement-based noise from the EEG signal in your experiments.
We thank the Reviewer for the valuable remark. We agree that EEG signals often require muscle artifact removal. There are specific advanced techniques for this, like the approach from mentioned work “Removal of movement artefact for mobile EEG analysis in sports exercises”. However, such methods commonly require additional biological signals to be recorded along with EEG – for example, electromyogram (EMG) or, like in this case, electrocardiogram (ECG). In the present study, we aimed to obtain results that would help develop passive brain-computer interfaces, so we tried to keep the experimental recording setup appropriate for such interfaces and as minimalistic as possible.
Additionally, muscle activity contamination is most notorious during intense physical activity such as sports exercises. In our experiment, the subject was sitting calmly at the table in a primarily static pose, so we believe muscle artifacts should not be very pronounced. However, any trials with substantial muscle artifact contamination were rejected and completely removed from the dataset.
Moreover, the conducted statistical analysis of the obtained data significantly reduces the effect of artifacts on the revealed effects.
We have added this discussion to the “EEG recording and processing” section.
The motivation for the used methods must be provided as there are many alternatives for each stage of your workflow available.
We believe that we have chosen the most appropriate and effective methods at each stage. In particular, for the spectral estimates, we used wavelet transform, which has proven itself in the analysis of bioelectric signals and has an optimal ratio of temporal and frequency resolution (see [Hramov A.E., Koronovskii A.A., Makarov Vă., Maksimenko V.A., Pavlov A.N., Sitnikova E.Yu. Wavelets in Neuroscience. Second edition. Springer Series in Synergetics. (2021)]). For statistical analysis, we used methods from the ANOVA family, which are the gold standard for solving such problems. To overcome the multiple comparisons problem in the statistical analysis of tomograms, we used the cluster-based permutation test, the necessity and sufficiency of which is shown in work [Maris E., Oostenveld R. Nonparametric statistical testing of EEG-and MEG-data // Journal of neuroscience methods. – 2007. – V. 164. – â„–. 1. – P. 177].
We have added the additional comments to the “Materials and Methods” section.
Line 193: present a motivation for using 2000 permutations. An appropriate reference can be used to support.
We have added the supporting references. This number of permutations is sufficient since it was empirically found that their increase does not change the results obtained for the considered problem.
Figure 3: the use of floating point numbers to enumerate conditions on the x-axis of the plots is confusing.
The floating-point numbers in this figure denote not the conditions but the wavelet power.
Improve the conclusions, present in-depth insights from your study. Current conclusions are too generic and do not follow from the numerical results of the experiments themselves. Use the main findings of your study to support your claim.
We thank the Reviewer for this important comment. We have improved the Conclusions section.
Check the language. There are some typos and misspellings.
Done.
Explain all abbreviations of their first use.
Done.
Reviewer 3 Report
Thanks for recommending me as a reviewer. The authors were use an EEG system to monitor and analyze the cortical activity of children and adults at a sensor level during cognitive tasks in the form of a Schulte table. If the authors complete the revision, the quality of the study will be further improved.
- The introduction section is well written. It may be helpful for readers to understand if the authors describe more specifically the theoretical background related to monitoring cortical activity of children and adults during cognitive task completion in the introduction section.
2. line 112-126: Authors should be more specific about the subject's characteristics in the Methods section. For example, authors may be more specific about the selection criteria, exclusion criteria, sampling, and general characteristics of subjects in this section.
3. line 170-187: In this study, the authors used the mixed-design ANOVA. If authors present the study design as a figure, it may help readers understand.
4. line 203: "The interaction 203 effect, number x table was insignificant: F(4.8, 43.1) = 4.38, p = 0.3." - As in other sentences, it would be better to indicate the significance level with 3 decimal places.
5. Authors should be more specific about the limitations of the study in the discussion section.
6. It may be helpful to readers if the authors add implications for future research in the Conclusion section.
Author Response
Thanks for recommending me as a reviewer. The authors were use an EEG system to monitor and analyze the cortical activity of children and adults at a sensor level during cognitive tasks in the form of a Schulte table. If the authors complete the revision, the quality of the study will be further improved.
We thank the Reviewer for the valuable comments, which we have considered in the revised version of the Manuscript.
Below are detailed answers to all of the Reviewer's comments.
- The introduction section is well written. It may be helpful for readers to understand if the authors describe more specifically the theoretical background related to monitoring cortical activity of children and adults during cognitive task completion in the introduction section.
We thank the Reviewer for the positive assessment of the introduction section and the comment. We have added the clarification regarding monitoring cortical activity (see p. 2, lines 50-54).
Mechanisms of interaction between the cognitive processes during mental tasks change with age. Uncovering the ways they change will substantially complement and advance our knowledge about brain development. To address this issue, we subjected children and adults to the Schulte Table (ST) with simultaneous recording of their brain electrical signals with EEG sensors and response time. Monitoring cortical activity during cognitive task completion with an EEG system allows obtaining objective information about the brain’s functioning and the occurring cognitive processes with a reasonable spatial and good time resolution [Michel, C.M.; Murray, M.M. Towards the utilization of EEG as a brain imaging tool. Neuroimage 2012, 61, 371; Xu, J.; Zhong, B. Review on portable EEG technology in educational research. Computers in Human Behavior 2018, 81, 340; Kannathal, N.; Acharya, U.R.; Lim, C.M.; Sadasivan, P. Characterization of EEG—a comparative study. Computer methods and Programs in Biomedicine 2005, 80, 17]. At the same time, electroencephalography is an easy-to-use and safe non-invasive technology that is especially important when working with children.
The applied motivation of the present study is the development of fundamental basics of functioning of passive brain-computer interfaces (BCIs) for monitoring and adjusting a human state in the process of learning and solving cognitive tasks [Hramov, A.E.; Maksimenko, V.A.; Pisarchik, A.N. Physical principles of brain-computer interfaces and their applications for rehabilitation, robotics and control of human brain states. Physics Reports 2021, 918, 1]. In this context, it is crucial to reveal age-related changes in cognitive processes and their interaction, allowing calibrating and optimizing BCIs for the corresponding age group. Such studies, in particular, are in demand for neuroeducation [Jamil, N.; Belkacem, A.N.; Ouhbi, S.; Lakas, A. Noninvasive Electroencephalography Equipment for Assistive, Adaptive, and Rehabilitative Brain–Computer Interfaces: A Systematic Literature Review. Sensors 2021, 21, 4754; Park, S.; Han, C.H.; Im, C.H. Design of wearable EEG devices specialized for passive brain–computer interface applications. Sensors 2020, 20, 4572].
- line 112-126: Authors should be more specific about the subject's characteristics in the Methods section. For example, authors may be more specific about the selection criteria, exclusion criteria, sampling, and general characteristics of subjects in this section.
We specified the characteristics of the subjects (see p. 3, lines 117-127). Particularly, the selection criteria were the following: no diagnosed diseases of the nervous system; no prescribed medications; non-smokers; right-handed; normal or corrected-to-normal visual acuity; amateur practitioners of physical exercises; never participated in this or similar experiments before; healthy life regime for a least 48 hours, including 8-hours night rest, prohibited alcohol consumption, limited caffeine consumption, and moderate physical activity. Exclusion criteria were the following: problems with completing the task; poor quality of recorded signals.
The age groups were selected based on the following considerations. Since we pay special attention to the development of BCIs for education, it was essential to select contrast age groups relevant to the performed task (ST). The first group consists of second-grade schoolchildren; the task is difficult for them because they are just beginning to master arithmetic operations. On the contrary, the volunteers from the second group, consisting of 1st-2nd-year students of IT specialty from Innopolis University, will efficiently complete ST since their arithmetic skills are predominantly formed at a high level.
- line 170-187: In this study, the authors used the mixed-design ANOVA. If authors present the study design as a figure, it may help readers understand.
We have considered this useful comment (see new Fig. 1 in the revised Manuscript).
- line 203: "The interaction 203 effect, number x table was insignificant: F(4.8, 43.1) = 4.38, p = 0.3." - As in other sentences, it would be better to indicate the significance level with 3 decimal places.
Corrected: p=0.301.
- Authors should be more specific about the limitations of the study in the discussion section.
The main limitation of this study is that we are considering two age groups (albeit representative ones). This limits our findings on age-related changes in cognitive processes to the age range from 7 to 20 years old. Nevertheless, we believe that this period of human life is the most important for improving many cognitive skills.
Considering a more diverse set of age groups is significant research that we plan to conduct in the future.
We have added these remarks at the end of the Discussion section.
- It may be helpful to readers if the authors add implications for future research in the Conclusion section.
The obtained fundamental results are essential for advancing our knowledge about brain development, including age-related changes in cognitive processes during mental tasks. From an applied perspective, the revealed effects are essential for developing passive brain-computer interfaces for monitoring and adjusting a human state in the process of learning and solving cognitive tasks [Hramov, A.E.; Maksimenko, V.A.; Pisarchik, A.N. Physical principles of brain-computer interfaces and their applications for rehabilitation, robotics and control of human brain states. Physics Reports 2021, 918, 1; Zander, T.O.; Kothe, C. Towards passive brain-computer interfaces: applying brain-computer interface technology to human-machine systems in general. Journal of neural engineering 2011, 8, 025005]. In particular, the identified patterns of neural activity associated with different strategies for solving tasks are important for developing algorithms that underlie the functioning of BCIs, which adapt to different age groups.
We have added these remarks at the end of the Conclusions section.
Round 2
Reviewer 1 Report
The authors clearly put a lot of work into improving the manuscript, which is remarkable considering the short deadlines.
Reviewer 2 Report
The paper has been improved sufficiently to be accepted for publication.